# Differential activation of NLRP3 inflammasome by *Acinetobacter baumannii* strains

**Fei-Ju Li, Lora Starrs, Anukriti Mathur, Hikari Ishii, Si Ming Man, Gaetan Burgio** *

Division of Immunology and Infectious Disease, The John Curtin School of Medical Research, The Australian National University, Canberra, ACT, Australia

* Gaetan.burgio@anu.edu.au

**Data Availability Statement:** All relevant data are within the paper and its Supporting Information files.

**Funding:** F.J.L is supported from the Taiwan-Australian National University Scholarship The

## Abstract

*Acinetobacter baumannii* is an emerging nosocomial, opportunistic pathogen with growing clinical significance globally. *A. baumannii* has an exceptional ability to rapidly develop drug resistance. It is frequently responsible for ventilator-associated pneumonia in clinical settings and inflammation resulting in severe sepsis. The inflammatory response is mediated by host pattern-recognition receptors and the inflammasomes. Inflammasome activation triggers inflammatory responses, including the secretion of the pro-inflammatory cytokines IL-1β and IL-18, the recruitment of innate immune effectors against *A. baumannii* infection, and the induction programmed cell death by pyroptosis. An important knowledge gap is how variation among clinical isolates affects the host's innate response and activation of the inflammasome during *A. baumannii* infection. In this study, we compared nine *A. baumannii* strains, including clinical locally-acquired isolates, in their ability to induce activation of the inflammasome and programmed cell death in primary macrophages, epithelial lung cell line and mice. We found a variation in survival outcomes of mice and bacterial dissemination in organs among three commercially available *A. baumannii* strains, likely due to the differences in virulence between strains. Interestingly, we found variability among *A. baumannii* strains in activation of the NLRP3 inflammasome, non-canonical Caspase-11 pathway, plasmatic secretion of the pro-inflammatory cytokine IL-1β and programmed cell death. Our study highlights the importance of utilising multiple bacterial strains and clinical isolates with different virulence to investigate the innate immune response to *A. baumannii* infection.

## Introduction

*Acinetobacter baumannii* (*A. baumannii*) is a Gram-negative nosocomial pathogen commonly causing pneumonia and sepsis and known to have a high prevalence of harbouring multi-drug resistance (MDR) genes [1]. Due to its rapid development of antibiotic resistance, *A. baumannii* is listed as one of the highly virulent 'ESKAPE' pathogens–a classification comprised of *A. baumannii*, *Enterococcus faecium*, *Staphylococcus aureus*, *Klebsiella pneumoniae*, *Pseudomonas aeruginosa*, and *Enterobacter* species [2]. Infections induced by these pathogens are responsible for life-threatening nosocomial infections in people who are most at risk, such as immuno-compromised patients or those critically ill in the intensive care unit (ICU) [3].

funders had no role in study design, data collection and analysis, decision to publish, or preparation of the manuscript.

**Competing interests:** The authors have declared that no competing interests exist

Despite a rising incidence of multi-drug resistant *A. baumannii* infections, the immune mechanisms resulting from the infection remain elusive. The first line of host defence involves innate immune pattern-recognition receptors (PRRs) sensing conserved structures of microbial organisms, called pathogen-associated molecular patterns (PAMPs) [4]. The PRRs' signalling cascades–triggered upon bacterial recognition–lead to TNFα and pro-IL-1β production, amplifying the inflammatory response. Pro-IL-1β is matured into the secretory form called IL-1β through the proteolytic activity of the cysteine protease Caspase-1 [5]. Mature IL-1β is an important pro-inflammatory cytokine, responsible for tightly regulating levels of inflammation in response to infection. It is activated by inflammasome sensor proteins, such as NLRP3 and NLRC4 [6, 7]. The role of the NLRP3 inflammasome has been characterized comprehensively for various diseases, including microbial infections. The assembly of the NLRP3 inflammasome following PRR signalling–and thus the activation of Caspase-1 and secretion of IL-1β–is termed the 'canonical inflammasome pathway' [8]. However, NLRP3 inflammasome activation may also be activated via a non-canonical inflammasome pathway involving Caspase-11 [8]. Caspase-11 is activated by direct binding to cytoplasmic Lipopolysaccharide (LPS); leading to Caspase-1-independent pyroptosis and NLRP3-Caspase-1-dependent release of IL-1β [4, 9].

In the limited studies on the immune response mounted against *A. baumannii*, there has been little investigation into the role of inflammasomes in the host innate immune defence. It has previously been described that during *A. baumannii* infection, Toll-like receptor (TLR) 4 and other PRRs are responsible for propagating the signal for cytokine/chemokine production and inflammasome activation [10–13]. This pathway leads to the rapid and robust downstream effector responses. Depletion of neutrophils [14, 15] or macrophages [16] revealed an important role for these innate immune cells during *A. baumannii* infection. In addition, using an intra-nasal infection model, Kang and colleagues suggested a reduced lung pathology in NLRP3-deficient, Caspase-1/11-deficient and IL-1-receptor-deficient mice, although the bacteria burden, recruitment of immune cells and production of inflammatory cytokines and chemokines were not altered in these mice [17]. In contrast, Dikshit and colleagues showed that although the NLRP3 inflammasome contributes to host defence against *A. baumannii* clinical isolates, it is not required to protect the host against a less virulent strain which is more sensitive to antibiotics [18].

Based on these findings, we hypothesised that inflammasome activation and programmed cell death are variable between the *A. baumannii* strains, correlating with bacterial virulence. To address this hypothesis, we examined the ability of three *A. baumannii* strains, frequently used in investigating bacterial biology, and six clinical isolates to induce pro-inflammatory cytokine production, inflammasome activation and programmed cell death. We indeed found a high variability in the survival of mice to infection with different strains of *A. baumannii* and variable bacterial dissemination in organs, likely due to the differences in strain virulence. Interestingly, we found variation among *A. baumannii* strains in the activation of the NLRP3 inflammasome non-canonical pathways via Caspase 11 activation and secretion of pro-inflammatory cytokines by the host. This potentially could correlate with the magnitude of programmed cell death observed and bacterial strain virulence. Holistically, our study underscores the importance of utilising multiple bacterial strains to investigate the host's innate immune response to *A. baumannii* infection.

## Method

### Bacteria culture

*Acinetobacter baumannii* strains were originally purchased from American Type Culture Collection (ATCC). Strain *A. baumannii* Bouvet and Grimont ATCC 19606, *A. baumannii* type

strain isolated from urine; strain *A. baumannii* Bouvet and Grimont (ATCC 17978), isolate from fatal meningitis of a four-month old infant; and strain *A. baumannii* ATCC BAA-1605 (short BAA-1605), a multi-drug resistant isolate from sputum of military personnel returning from Afghanistan entering a Canadian hospital. Additional clinical isolates including 834321, 820642, 916085, 938408, 834625 and 914394 were characterised and provided by the Canberra Hospital and are listed in **S1 Table**. Bacteria were grown in 5 ml Trypticase soy broth (TSB) at 37˚C prior to single colony isolation for 18 hours at 37˚C with shaking at 220 rotations per minute (rpm). These single colonies were expanded and frozen at -80˚C in 30% glycerol. The concentration of the glycerol stocks for each strain was calculated by thawing three glycerol stocks and culturing each in 15 ml of TSB at 37˚C with shaking at 220 rotations per minute (rpm) for 18 hours, before enumeration on trypticase soy agar (TSA) plates for 16–18 hours at 37˚C, and the average CFU/ml for each strain was calculated. To determine whether the isolates were from *A. baumannii*, bacterial selection was performed on TSA and MacConkey agar media, gram staining. Isolates were identified by matrix-assisted laser desorption ionisation–time of flight mass spectrometry (Bruker). Susceptibility testing was performed using Vitek MS (bioMerieux) and interpretated using European Committee on Anitmicrobial Susceptibility Testing (EUCAST) breakpoints and determine using a Vitek device (Biomerieux). For experiments herein, controls were set-up as 'mock' infection, that is, treated under identical conditions with the exception of adding bacteria.

For growth curve, all bacteria strains were grown for a total of 18 hours in TSB at 37˚C with shaking at 220 rpm. CFU/ml concentration was determined by sampling 100 µl every two hours post inoculation. Colonies were enumerated on TSA, cultured for 16–18 hours at 37˚C.

## Cell culture

**a. A549 cell line.**   The human carcinomic, lung epithelial line A549 was purchased from ATCC, and maintained in complete media consisting of Ham's F-12K Medium (21127022, Gibco ThermoFisher) with 10% foetal bovine serum (FBS, F8192, Sigma) and supplemented with penicillin/streptomycin/glutamine (PSG, 10378016, Gibco ThermoFisher). Cells were maintained at 37˚C in 5% CO2 with >90% relative humidity. Cells were seeded in antibiotic-free media at a concentration of $1 \times 10^5$ cells per well in 96-well plates the night prior to infection.

**b. Bone Marrow Derived Macrophages (BMDMs).**   Primary BMDMs were isolated from both mouse femur and tibia under sterile conditions and in accordance with protocols mentioned in 'Animals'. Briefly, bone marrow was flushed with RPMI with 2x penicillin/streptomycin and filtered through a 70um strainer (Cat No.:352350, BD Falcon). BMDM were isolated from red blood cells (RBC) through incubation of bone marrow with RBC lysis buffer (154.4mM $NH_4Cl$, 9.99mM $KHCO_3$ and 0.1267nM EDTA) and centrifugation at 1,500 rpm for 5 minutes at room temperature. Cells were washed with RPMI with 2x penicillin/streptomycin and re-centrifuged at 1,500 rpm for 5 minutes at room temperature. Cells were seeded in a sterile 10-cm dish at a density of $5x10^6$ per dish in complete BMDM differentiation media consisting of RPMI-1640 media (22400–089, Gibco ThermoFisher) supplemented with 10% FBS, 1% penicillin and streptomycin (10378016, Gibco ThermoFisher) and 10 ng/ml mouse GM-CSF (Cat. No.: 130-095-739, Miltenyi Biotech). Cultures were maintained at 37˚C in 5% CO2 with >90% relative humidity. Differentiation media was topped-up on day 3, and BMDMs were harvested on day 6 or 7 depending on confluency. BMDMs were harvested in 5 µM EDTA/RPMI and centrifuged at 1,500 rpm for 5 minutes at room temperature and then seeded in antibiotic-free media at a concentration of $1 \times 10^5$ cells per well in 96-well plates and incubated overnight at 37˚C in 5% CO2 with >90% relative humidity prior to infection.

## Quantification of cell death

**a. Trypan blue.** A549 cells were seeded into sterile glass bottom 24-well plate (Cat. No.: P24-1.5HN, Cellvis) at $4x10^5$ cells/well in antibiotic-free and serum-free media prior to *A. baumannii* inoculation. Cells were treated with different strains of *A. baumannii* at a multiplicity of infection (m.o.i.) of 80 for 24 hours (final volume 1 ml/well) at 37˚C with 5% $CO_2$ with >90% relative humidity. Cells were washed twice with sterile 1xPBS before staining with 2% trypan blue in 1xPBS at room temperature for 5–10 minutes. Post staining, cells were wash with sterile 1xPBS prior to fixation with 4% paraformaldehyde (PFA) (Cat. No.: 420801, BioLegend) at room temperature for at least one hour. PFA was carefully removed and 1xPBS was added to the wells for imaging. Cells were imaged using Zeiss Axio Observer under 40x objective. Five random fields were chosen per well and quantified using Image J with colour deconvolution plugin (version 1.64r).

**b. Microscope.** BMDMs were seeded into sterile glass bottom 24-well plate (Cat. No.: P24-1.5HN, Cellvis) at $4x10^5$ cells/well prior to *A. baumannii* inoculation. *A. baumannii* strains were prepared at m.o.i. of 10 and incubated with the cells for 24 hours (final volume 1 ml/well) at 37˚C 5% $CO_2$ and >90% relative humidity. Post infection, cells were washed twice with sterile 1xPBS before staining with Zombie Aqua (1:1000, 100 µl/sample, Cat. No.: 423101, BioLegend) and Hoechst 33342 (1:50, 100 µl/sample, Cat. No.: 62249, Thermo Fisher Scientific) for 30 minutes in the dark at room temperature. Post staining, cells were washed twice and fixed in 4% PFA (Cat. No.: 420801, BioLegend) for 30 minutes in the dark, at room temperature. PFA was carefully removed and 1xPBS was added to the wells for imaging. Samples were imaged using Zeiss Axio Observer with an epifluorescence attachment and a digital camera. Five random fields were taken per well and quantified using Image J for mean staining area per channel (ver 1.64r).

## Animals

8-12-week-old mice were used for all mouse infection experiments and mice were between 8–16 weeks of age for BMDM isolation. C57BL/6J/[ANU] mice were purchased from the Australian Phenomics Facility, the John Curtin School of Medical Research (Canberra, Australia). *Caspase-11*[-/-] mice were sourced from the Jackson Laboratory [19]. All mice were maintained under specific pathogen-free conditions. All animal studies were performed in accordance with the National Health and Medical Research Council code for the care and use of animals under the Protocol Numbers A2018-08 and A2021-14 approved by The Australian National University Animal Experimentation Ethics Committee.

## Mouse infection

**a. Bacterial preparation and inoculation.** C57BL/6J/[ANU] mice were infected with $2x10^7$ CFU/mouse *A. baumannii* strains (ATCC 19606, ATCC 17978 or ATCC BAA-1605) intraperitoneally. Briefly, single bacteria colony was grown for 16–18 hours in TSB at 37˚C, shaking at 220-250rpm. Next morning the cultures were refreshed in TSB (1:5) and incubated at 37˚C with shaking for another 2 hours. The bacteria were then centrifuged at 2,300 rpm for 30 minutes and washed with sterile 1xPBS. The bacteria were resuspended at a concentration of $2x10^7$ CFU/200 µl/mouse. Actual inoculum concentrations were determined by plating serial dilutions on trypticase soy agar (TSA) plates.

**b. Mouse monitoring and survival.** Infected mice were monitored at 4, 8, 16, 20, and 28h post infection. The infection was allowed to proceed for 28 hours before the mice were humanely euthanized, and the organs were harvested for bacterial burden quantification. Mice were monitored and scored based on condition, as assessed by factors including activity levels

(spontaneous activity without stimulus: 0 slowing in movements: 2 unresponsive to stimuli: 5.), grooming levels (smooth: 1 vs. rough coat: 5), and hydration levels (normal drinking and eating:0 dehydration: 5). The humane endpoint of an experiment for each mouse was determined based on condition in accordance with the animal ethics protocol and the National Health and Medical Research Council code for the care and use of animals. Special housing conditions were taken, such as the introduction of igloos, additional bedding, liquid food on the floor to minimize suffering and distress. Animals with scores > 5 were immediately humanely euthanized. Animals were also humanely euthanized within 20 hours post-inoculation due to severe sepsis following the infection. No animals were found dead. Tissues were collected to determine the bacterial burden and the immune response. The rest of the cohort was humanely euthanized at the experimental endpoint (28 hours) for tissue collection.

**c. Quantification of bacterial burden.** For enumeration of the bacteria burden in the lungs, liver, spleen and kidneys, mice were humanely euthanised between 8-20-hour post infection. Organs were aseptically removed and weighed, prior to homogenisation in sterile 1xPBS. Approximately 20 to 50 mg of each organ were and filtered through a 70 μm nylon mesh cell strainer in 1 ml of sterile 1xPBS. Serial dilutions were then plated on TSA and incubated at 37˚C for 16–18 hour.

For enumeration of bacterial burden in blood, blood was harvested in EDTA-containing tubes via post-euthanasia cardia-puncture. Serial dilutions of blood were plated on TSA plates to quantify bacterial load.

**Real-time reverse transcriptase polymerase chain reaction (real-time RT-PCR).** RNA was extracted from BMDMs using TRIzol (15596018, ThermoFisher Scientific). The isolated RNA was converted into cDNA using the High-Capacity cDNA Reverse Transcription Kit (4368813, ThermoFisher). RT-PCR was performed on an ABI StepOnePlus System PCR instrument with SYBR Green Real Time PCR Master Mixes (1725270, Bio-rad). Primer sequences can be found in **S2 Table**. Real-time RT-PCR data and fold change (compared to mock) was calculated using the comparative $2^{-\Delta\Delta CT}$ method [20] with *Gapdh* as a housekeeping gene. Relative expression was displayed as fold-changes.

## Enzyme-linked immunosorbent assay (ELISA)

For cytokine measurement, plasma was collected after 8–20 hour for analysis by ELISA. Serial diluted standard with known concentration was included in each plate. Cytokine concentrations from BMDMs were measured using TNFα (88-7324-77, Invitrogen), or IL-1β (88-7013-77, Invitrogen) ELISA kits according to the manufacturers' instructions. All plates were measure using TECAN Infinite® 200 Pro (Tecan, Männedorf, Switzerland), with wavelength set at 450 nm.

**Lactic dehydrogenase (LDH) assay.** LDH released by cells was measured using a Cyto-Tox 96 Non-Radioactive Cytotoxicity Assay, performed according to the manufacturer's instructions (Cat. No.: G1780, Promega).

## Western blotting

**a. Sample preparation.** Post infection, BMDMs and supernatant sample were lysed in Radioimmunoprecipitation assay buffer (RIPA) lysis buffer supplemented with protease inhibitors, i.e., Complete Protease Inhibitor Cocktail Tablets (Cat No.: 04693132001, Roche) to prevent sample degradation. Samples were heated at 95˚C for 5 minutes with 6x Laemmli buffer containing Sodium Sodecyl Sulfate (SDS) and 100 mM Dithiothreitol (DTT) for 5 minutes before storing at -80˚C.

**b. Sodium Dodecyl Sulfate Polyacrylamide Gel Electrophoresis (SDS-PAGE).** Samples were thawed on ice and then denatured at 95˚C for 10 minutes. Each sample was loaded on an individual lane of a 4–15% gradient SDS-PAGE gel (Cat No.: 456–1086, Bio-Rad) and run under reducing conditions with a constant voltage of 200 volts for approximately 25 minutes until the dye front reached the end of the gel. The resolved proteins in the SDS-PAGE gel were then transferred to a 0.45 μm Polyvinylidene fluoride (PVDF) membrane (Cat No.: 1620115, Bio-Rad) by electroblotting. An electric current of with 400 mA was applied to the apparatus for 1.5 hours at 4˚C. Following the transfer, the membrane was blocked with 5% (w/v) skim milk in PBS for 1 hour at room temperature to prevent non-specific binding of Immunoglobulins (Ig).

**c. Detection of protein of interest.** Post-blocking the PVDF membrane was incubated with primary anti-Caspase-1 (1:1000, Cat. No.: 106–42020, Adipogen), Caspase-11 (1:1000, Cat. No.: NB120-10454, Novusbio), or Glyceraldehyde 3-phosphate dehydrogenase (GAPDH) (1:1000, Cat No.: MAB374, Merck Millipore), diluted in 1% (w/v) skim milk in PBST (1xPBS with 0.1% Tween-20) overnight at 4˚C with gentle rocking. PVDF membranes were then incubated with species-specific horseradish peroxidase-conjugated secondary antibodies (1:5000) for 1 hour at room temperature with gentle rocking. Immunoreactive proteins were detected by applying ECL Western blotting Detection Reagent (Cat No.: 1705060, Bio-Rad) or Super-Signal™ West Femto Maximum Sensitivity Substrate (Cat. No.: 34096, Thermo Fisher Scientific). The results were collected using ChemiDoc™Touch Imaging System (BioRad).

**Omp38 protein alignment.** Omp38 FASTA format sequences from *A. baumannii* strains were obtained from the National Center for Biotechnology information (NCBI) with the respective accession IDs; ATCC BAA 1605 (EPS78178), ATCC 19606 (KFC05096) and ATCC 17978 (CAA6833801). Sequences were aligned using Clustal Omega [21] and aligned to the secondary *A. baumannii* Omp38 structure (Protein Data Bank 3TD3) using ESPript 3.0 [22].

## Statistical analysis

The GraphPad Prism 9.0 software was used for data analysis. Data are shown as mean ± s.e.m. Statistical significance was determined by t-tests (two-tailed) for two groups or one-way analysis of variance (with Dunnett's multiple-comparisons test) for three or more groups. Survival curves were compared using the log-rank test. $P < 0.05$ was considered statistically significant.

## Results

### Multi-drug resistant *A. baumannii* ATCC BAA-1605 induced the highest lethality and bacterial burden

To assess the virulence of the *A. baumannii* strains ATCC 19606, 17978 and 1605, we examined tissue-specific bacterial burden and lethality in mice. Using the same dose ($2x10^7$ CFU/mouse via intraperitoneal inoculation), the strain ATCC 17978 induced no lethality (**Fig 1A**), low bacterial load in blood (**Fig 1B**), and low colonisation level ($< 100$ CFU/mg) in the lung, liver, spleen, and kidney 16–20 hours post inoculation (**Fig 1C**). In contrast, the MDR strain ATCC BAA-1605 induced lethality in 62% of the mice (**Fig 1A**), with a high bacterial burden in the blood (**Fig 1B**), lung, liver, spleen or kidney (**Fig 1C**)**,** while the virulence and colonisation rate of the ATCC 19606 strain was intermediate between the ATCC 17978 and the ATCC BAA-1605 strains. Next, we compared the level of pro-inflammatory cytokines in the plasma of infected mice. Both the strain ATCC 19606 and the MDR strain ATCC BAA-1605 induced higher plasmatic IL-18 and IL-1β secretion level compared to the avirulent ATCC 17978 (**Fig 1D**). Together, these results suggest these *A. baumannii* strains differentially colonise, trigger inflammation and pro-inflammatory cytokine release, leading to differences in survival.

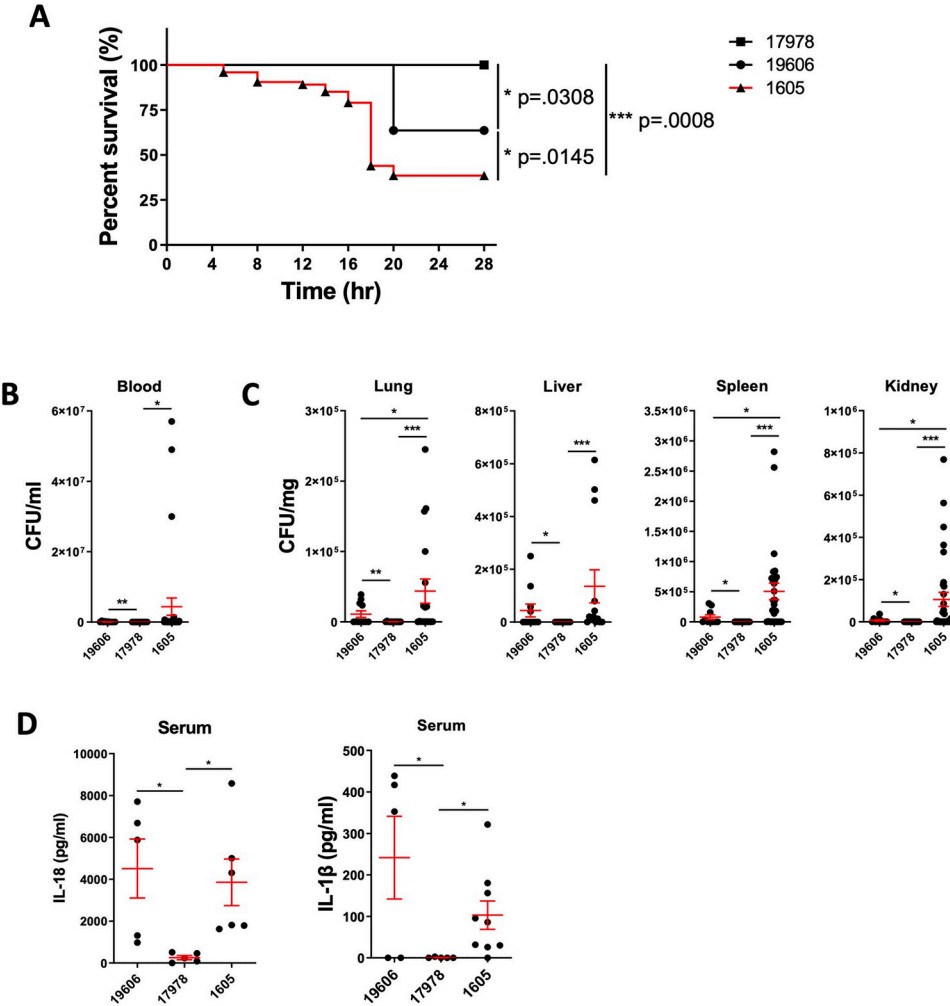

**Fig 1. Differential virulence of *A. baumannii* strains in a mouse infection model.** (A) The survival of C57BL/6 mice following infection by three different strains of *A. baumannii* (i.p. $2\times10^7$ CFU/mouse). The level of (B) bacteraemia and (C) bacteria dissemination to different organs at 16–20 hours post infection was quantified. (D) Serum levels of IL-18 and IL-1β in C57BL/6 mice post 16–20 hours intraperitoneal *A. baumannii* infection. Data were collected from at least three independent experiments, n = 11 (ATCC 19606 and 17978) and n = 13 (ATCC BAA 1605). (b, c, d) data are shown as mean ± SEM. *, Log-rank survival and one-way ANOVA statistical tests were performed. P < 0.05, **, P < 0.01, ***, P < 0.001.

## Differential upregulation of inflammasome components by different *A. baumannii* strains

We next determined which inflammasome is responsible for the maturation and release of pro-inflammatory cytokines IL-1β and IL-18. We checked the expression of inflammasome sensors via qRT-PCR on bone marrow derived macrophages (BMDMs) over 24 hours of infection. For three *A. baumannii* strains, induction of NLRP3, Caspase-1, LDH release and cell death (**S1A and S1B Fig**) was observed in BMDMs six hours post inoculation, which was sustained until 12 hours (**Fig 2A**). Interestingly, we observed a prolonged 8.4-fold induction of Caspase-11 24 hours post inoculation in BMDMs inoculated with the strain ATCC 17978 compared to ATCC 19606 or MDR strain ATCC BAA-1605. The MDR ATCC BAA-1605 strain induced a modest two-fold increase NLRC4 expression 24 hours post inoculation

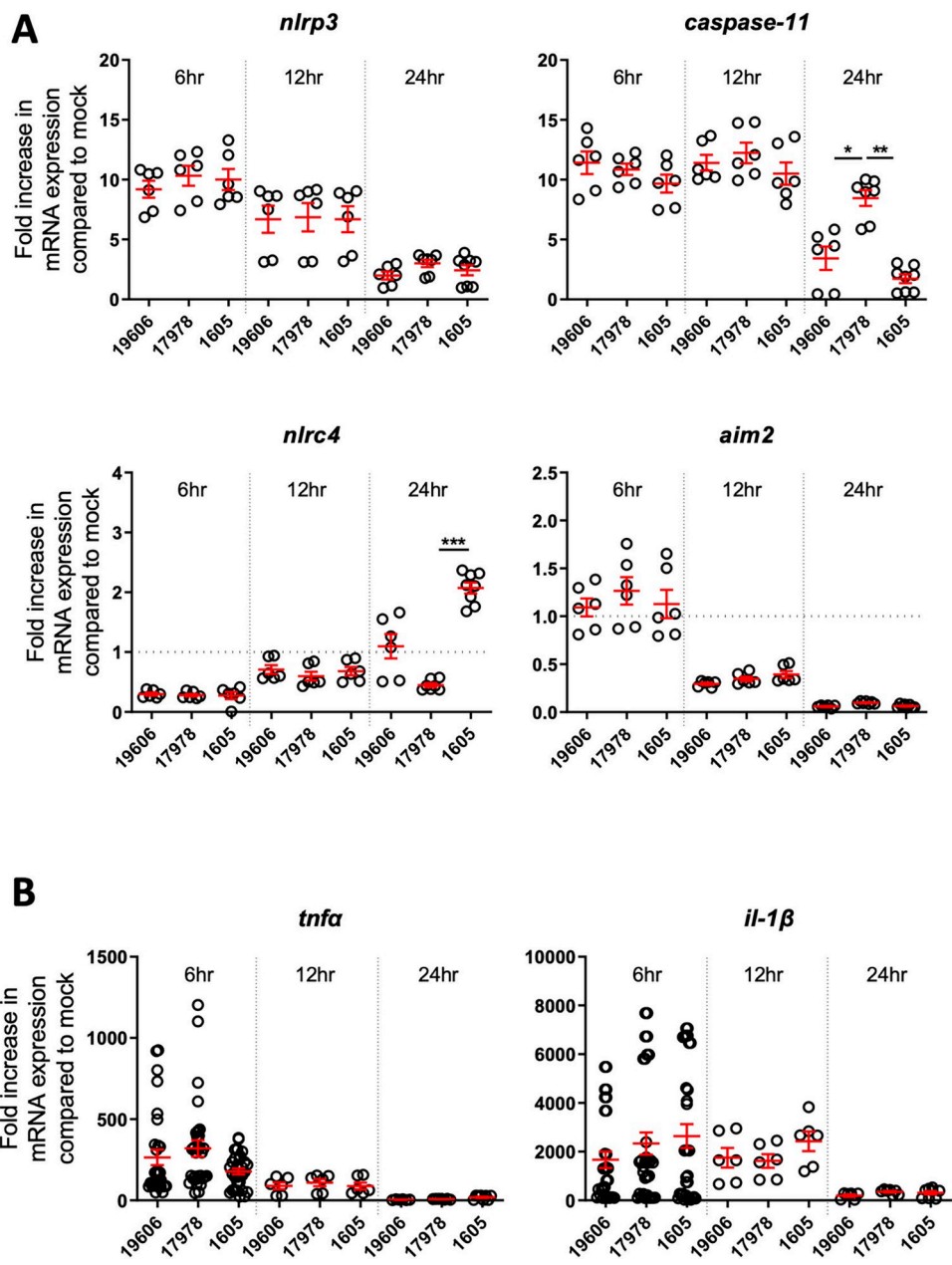

**Fig 2. Mouse BMDMs were infected with m.o.i. 10 of live *A. baumannii* and analysed at the timepoints indicated.**
(A) qRT-PCR transcript levels of *Nlrc4*, *Aim2*, *Nlrp3* and *Caspase-11* during infection compared to mock infection.
N = 6 per strain (B) Tnfα and IL-1β, produced by wild-type mouse BMDMs between 6–24 hours post-infection
(lysate). N = 12 per strain (6 hours post inoculation) and n = 6 per strain at 12 and 24 hours post inoculation. Unpaired
t-tests were performed. mean ± SEM. *, P < 0.05, **, P < 0.01, ***, P < 0.001.

compared to mock (where a fold induction of 1 is equivalent to mock treatment) (**Fig 2A**). We
noted no induction of the gene encoding AIM2 by any of the *A. baumannii* strains tested. We
found no difference in the pro-inflammatory cytokines TNFα or IL-1β levels among the three
strains from 6 to 24 hours post inoculation in BMDMs by qRT-PCR (**Fig 2B**) or ELISA (**S2
Fig**). Overall, our findings suggest a consistent and preferential upregulation of NLRP3 and
Caspase-11 in response to several strains of *A. baumannii*.

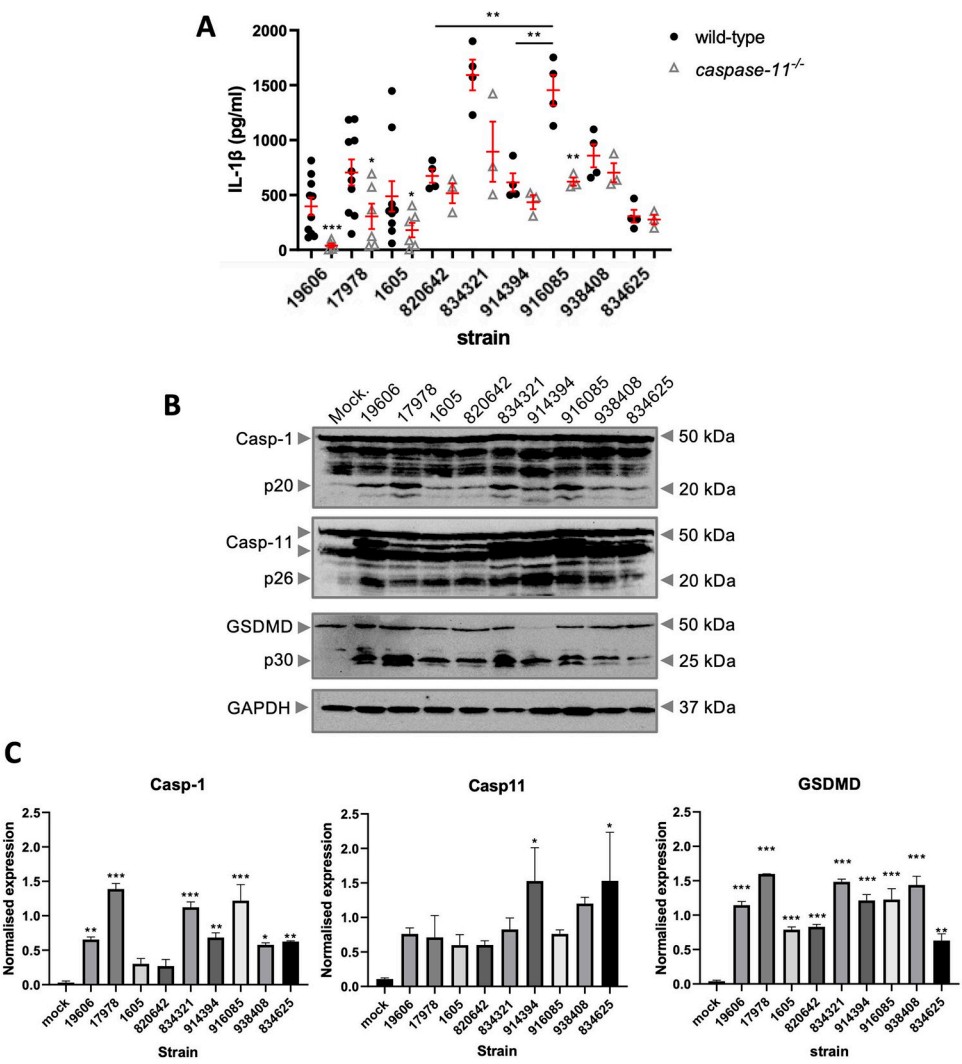

**Fig 3. Differential activation of the inflammasome components by different *A. baumannii* strains.** (A) ELISA of IL-1β levels produced by mouse BMDM after infection m.o.i. 10 of *A. baumannii* infection for 12 hours. mean ± SEM. N = 8 per ATCC strains and n = 4 for the clinical isolates (B) Representative western blot image of caspase-1, caspase-11, and GSDMD post infection with different *A. baumannii* strains. (C) Western blot densitometry quantification normalised to GAPDH and compared to mock. n = 2 per isolates. Unpaired t-test statistical tests were performed mean ± SEM. *, P < 0.05, **, P < 0.01, ***, P < 0.001.

We next hypothesised the magnitude and type of inflammatory response to *A. baumannii* is strain-dependent. BMDMs were infected with six *A. baumannii* clinical strains with various degree of resistance to antibiotics, including Carbapenem (**S1 Table**). We firstly quantified their growth rates (**S3 Fig**) and found only the strain ATCC 19606 displayed a lower growth rate. We next quantified Caspase-1, Caspase-11 and Gasdermin D (GSDMD) cleavage by immunoblotting in wild-type (WT) BMDMs 12 hours post infection. We performed further analysis by measuring the associated cytokine IL-1β level in the supernatant of WT and *Caspase 11⁻/⁻* BMDMs 12 hours post infection. We noted a high variation in secreted IL-1β between all the clinical strains tested. The multidrug resistant strains 834321 and 916085 seem to induce the highest levels of IL-1β secretion in WT BMDMs, but not in *Caspase 11⁻/⁻* BMDMs (**Fig 3A**). Unsurprisingly, infection with all of the strains resulted in Caspase-1

cleavage. Yet interestingly, there is large variation in the activation of Caspase-11. Only strains 914394 and 834625 induced highest activation of Caspase-11 (**Fig 3B**). Together it suggests high variability in the inflammasome activation and IL-1β between and among the tested *A. baumannii* strains.

**A. baumannii infection induces host programmed cell death.** The activation of GSDMD and the release of IL-1β suggest that *A. baumannii* strains were likely to induce pyroptosis in BMDMs. We therefore quantified the level of cell death in BMDMs using the cell membrane permeability dye Zombie aqua, which detects cells with damaged membranes. We observed a variation in the level of cell death between BMDMs infected with different strains, with ATCC 17978 being the lowest (**Fig 4A and 4B**), which might explain its low *in vivo* virulence (**Fig 1**). To investigate this further, we next determined whether this trend would be observed for a cell type other than BMDMs. As *A. baumannii* preferentially invades lung epithelium [23], we quantified cell death in the human A549 lung epithelial cell line. We infected A549 cells with *A. baumannii* and analysed using trypan blue uptake assay (**Fig 4C**). Based on our cell membrane permeability assays, the strain ATCC BAA-1605 induced the highest level of cell death when compared to ATCC 17978 and ATCC 19606 strains (**Fig 4A and 4B**). We next postulated we would observe a high variation in cell death in response to infection by clinical strains. Amongst all strains tested, Trypan blue staining revealed that the clinical strain 916085 induced the highest level of cell death in A549 cells (**Fig 4C**). Overall, it suggests a high variability of the *A. baumannii* strains to induce cell death.

## Discussion

*Acinetobacter baumannii* is a Gram-negative bacterium that causes opportunistic infections in humans. Despite its clinical significance, there is a lack of studies evaluating strain-dependency in the induction of inflammation and cell death responses. We examined strain-to-strain variation in virulence, pro-inflammatory cytokine secretion, inflammasome activation and programmed cell death. We observed a notable difference in bacterial growth, virulence, pro-inflammatory cytokines, Caspase-11 activation and programmed cell death in three commercially available *A. baumannii* strains ATCC 17978, 19606 and BAA-1605 commonly used to investigate *A. baumannii* pathogen biology and virulence. Here, we show that MDR ATCC BAA-1605 is more virulent than the ATCC 19606 and 17978 strains. Consistent with a previously published study [24], we found that *A. baumannii* is rapidly disseminated via the blood to peripheral organs, including the lung and spleen. Additionally, we show that the least virulent strain 17978 induced minimal systemic inflammatory responses, suggesting the acute lethality of other strains might be dependent on the degree of inflammatory response and pro-inflammatory cytokine releases.

Given the differential pro-inflammatory cytokine release and programmed cell death induced by different *A. baumannii* strains in mice, we wondered which inflammasome sensor and pathway was activated. Consistent with previous studies [17, 18, 25], we found an induction of NLRP3 and Caspase-11 inflammasome components early in the infection. Although it does not contribute to the differential cytokine release, the extended induction of Caspase-11 by the least virulent strain 17978 and the delayed induction of NLRC4 by MDR ATCC BAA-1605 correlated with cell death. We confirmed this observation using six additional clinical isolates which exhibit various degrees of antibiotic resistance. We have titrated different m.o.i. versus IL-1β secretion and noted a lack of dose dependency (**S2 Fig**). We reasoned that the activation of inflammasomes largely depended on live *A. baumannii* activity rather than the presence of virulence factors.

We examined programmed cell death in BMDMs and A549 human epithelial cell line using three commercial *A. baumannii* strains and six additional clinical isolates with various

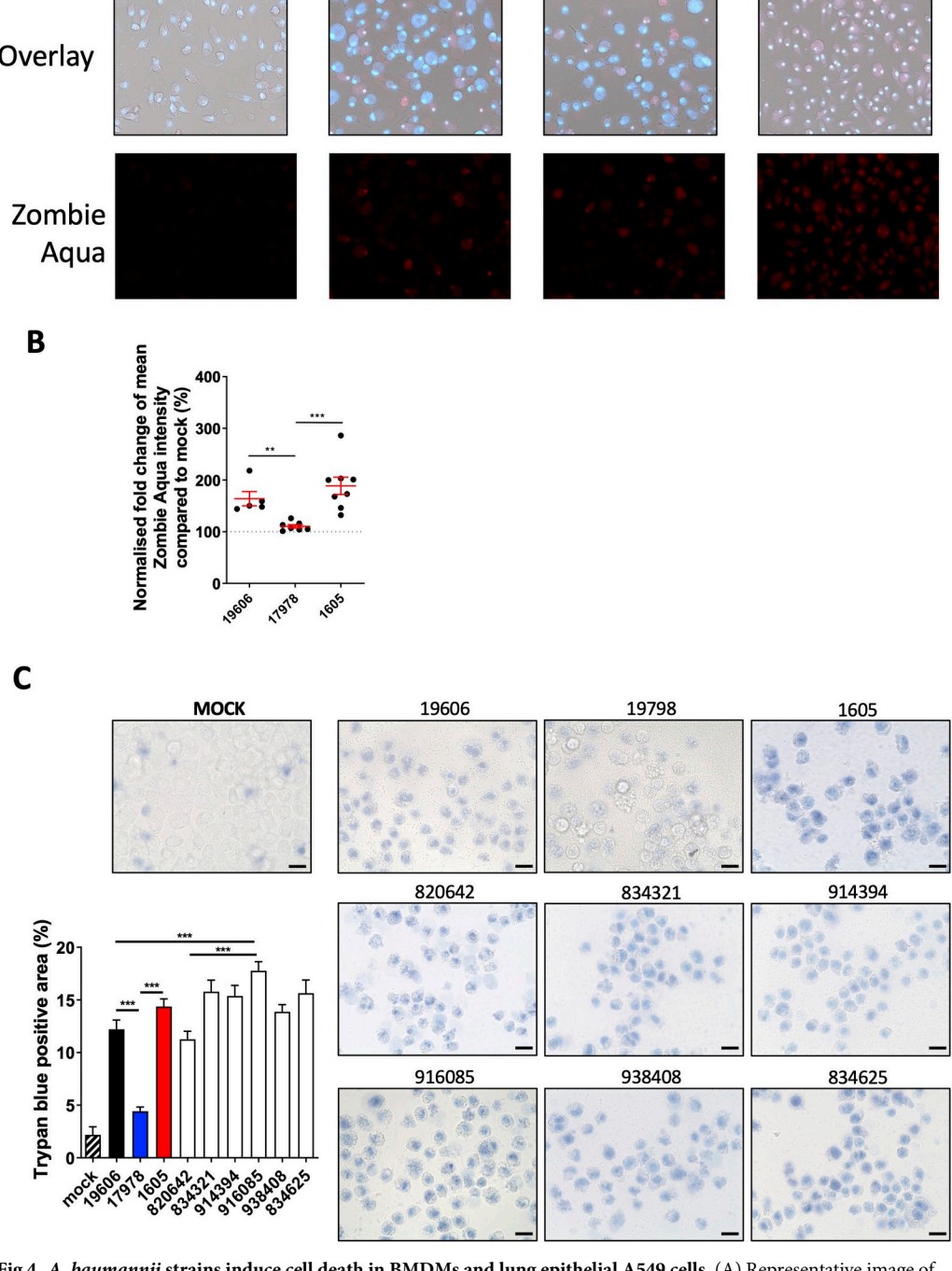

**Fig 4. *A. baumannii* strains induce cell death in BMDMs and lung epithelial A549 cells.** (A) Representative image of wild-type mouse BMDM cell death post m.o.i. 10 of A. *baumannii* infection for 24 hours with Hoechst (blue) and Zombie aqua (red) staining. Brightfield staining was included (top panel) (B) Densitometry quantification of the Hoechst and Zombie aqua staining from 5 random field per samples. N = 3 biological replicates per sample. Two-tailed t-tests was performed (C) Representative image and densitometry quantification of A549 cell death post A. *baumannii* infection at m.o.i. 80 for 24 hours, scale bar: 20 μm. N = 4 per strain. Unpaired t-tests were performed. All data are shown as mean ± SEM. *, P < 0.05, **, P < 0.01, ***, P < 0.001.

antibiotic resistance profiles. We found that the multidrug resistant clinical isolate 916085 induced the highest cell death. Interestingly, we have found a trend in relation to the variation in cell death versus inflammasome activation and pro-inflammatory cytokine secretion in our commercial and clinical strains. This suggests *A. baumannii* virulence factors are potentially modulating inflammasome activation and programmed cell death. Previous work demonstrated the role of virulence factors such as the IAV BP1-F2 [26], the UPEC alpha-hemolysin [27] and toxins of *Bacillus cereus* [28, 29] in determining NLRP3 activation and severe pathogenicity. Recently, the virulence-related Omp38 was identified as an activator of NLRP3 inflammasome via Caspase-1 in *A. baumannii* infections [30]. We performed a multiple sequence alignment of the Omp38 protein sequence between the strains ATCC 17978, 19696 and BAA 1605 and found no differences in the OmpA-like domain (**S4 Fig**). Further analyses would assist with detailed characterisation of additional virulence factors in *A. baumannii* which may be responsible for Caspase-11 and NLRP3 activation leading to pyroptosis.

In conclusion we found the MDR ATCC BAA-1605 was the most virulent strain both *in vitro* and *in vivo* and showed the highest cytotoxicity from our commercially available strains. Further, we also identified, from our assessment of the commercially available strains, that the induction of different inflammasome transcripts does not directly equate to the level of inflammation in BMDMs or cell lines. This was further supported by our assessment of clinical isolates, which showed high levels of variation in the pro-inflammatory cytokine response and programmed cell death *in vitro*. Additionally, due to a lack of dose-dependency *in vitro*, the activation of inflammasome was thought to be largely dependent on bacterial virulence. We finally propose that the combination of *in vitro* cell death and *in vitro* Caspase-11-dependency may be harnessed as a screening tool to select for more virulent strains to use *in vivo*.

## Supporting information

**S1 Fig. Assessment of inflammation and cell death from 6 to 24 hours post *A. baumannii* infection in wild type mice.** (A) LDH assay of mouse BMDM post *A. baumannii* infection. mean ± SEM. N = 3 per timepoint. (B) Zombie aqua quantification of mouse BMDM death post *A. baumannii* infection (strain ATCC BAA 1605). mean ± SEM. N = 4.
(TIF)

**S2 Fig. Pro-inflammatory cytokine secretion at 12 to 24 hours post-inoculation in BMDMs.** Mouse BMDMs were infected with indicated m.o.i. of live *A. baumannii* and analysed. ELISA of pro-inflammatory cytokines levels of TNFα and IL-1β produced by wild-type mouse BMDM after 12 and 24 hours of infection, n = 8 per strain. mean ± SEM.
(TIF)

**S3 Fig. Bacterial growth assay.** *A. baumannii* isolates growth rate (CFU/ml) up to 30 hours post inoculation. N = 7 per strain. All data are shown as mean ± SEM.
(TIF)

**S4 Fig. Omp38 sequence alignment from ATCC BAA 1605 (EPS78178), ATCC 19606 (KFC05096) and ATCC 17978 (CAA6833801).** Protein sequences were aligned to Omp38 secondary structure and its OmpA-like domain (residues 221–339). The OmpA-like domain is fully conserved in all aligned protein sequences.
(TIF)

**S1 Table. ATCC and clinical *A. baumannii* strain isolation sites and antibiotic resistance profiles.**
(DOCX)

**S2 Table. Mouse primers used for qRT-PCR.**
(DOCX)

# Acknowledgments

We would like to thank Mr Mick Devoy and Dr Harpreet Vora, the Biomolecular resource facility and the Australian Phenomics Facility for technical assistance. The authors would like to acknowledge Dr Karina Kennedy, Dr Susan Bradbury for providing the clinical *A. baumannii* strains. F.J.L is supported from the Taiwan-Australian National University Scholarship. H.I is supported from the TOBITATE young ambassador program.

# Author Contributions

**Conceptualization:** Gaetan Burgio.

**Data curation:** Lora Starrs.

**Formal analysis:** Lora Starrs, Si Ming Man, Gaetan Burgio.

**Funding acquisition:** Gaetan Burgio.

**Investigation:** Fei-Ju Li, Lora Starrs, Anukriti Mathur, Hikari Ishii.

**Methodology:** Fei-Ju Li, Lora Starrs.

**Resources:** Fei-Ju Li, Lora Starrs, Anukriti Mathur, Hikari Ishii, Si Ming Man, Gaetan Burgio.

**Writing – original draft:** Fei-Ju Li, Gaetan Burgio.

**Writing – review & editing:** Fei-Ju Li, Lora Starrs, Anukriti Mathur, Si Ming Man, Gaetan Burgio.

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
