## [Decision Letter · Decision Letter 0]

8 Aug 2022

PONE-D-22-16629Differential activation of NLRP3 inflammasome by Acinetobacter baumannii strainsPLOS ONE

Dear Dr. Burgio,

Thank you for submitting your manuscript to PLOS ONE. After careful consideration, we feel that it has merit but does not fully meet PLOS ONE’s publication criteria as it currently stands. Therefore, we invite you to submit a revised version of the manuscript that addresses the points raised during the review process.

The results presented are very interesting. The study will be strengthened if additional evidence can be incorporated by conducting a few experiments suggested by one of the authors.

Quantification of the immunoblotting is essential.

We look forward to receiving your revised manuscript.

Kind regards,

Suresh Yenugu

Academic Editor

PLOS ONE

Journal Requirements:

2. Please report additional details in your Methods section regarding animal care and use for the survival study, as per our editorial guidelines (http://journals.plos.org/plosone/s/submission-guidelines#loc-humane-endpoints).

For easy reference, we have attached a checklist that may be relevant for your submission. Please complete all items on the checklist at the following link: http://journals.plos.org/plosone/s/file?id=bb1d/plos-one-humane-endpoints-checklist.docx

Please upload the completed checklist as file type “Other” when resubmitting your manuscript. This document is for internal journal use only and will not be published if your article is accepted. We very much appreciate your attention to these requests and support of improved reporting standards in PLOS ONE submissions.

5. PLOS ONE now requires that submissions reporting blots or gels include original, uncropped blot/gel image data as a supplement or in a public repository. This is in addition to complying with our image preparation guidelines described at https://journals.plos.org/plosone/s/figures#loc-blot-and-gel-reporting-requirements. These requirements apply both to the main figures and to cropped blot/gel images included in Supporting Information. If the manuscript is positively reviewed, we will ask the authors to provide any missing raw image data for blot/gel results when they submit their first revision. As part of your review, please ensure that figures reporting blot or gel images comply with the journal’s image preparation guidelines and that the original data are provided following the journal’s request.  If you have any questions or concerns about blot/gel figures or data for this submission, please email us at plosone@plos.org before issuing a decision letter.

Reviewers' comments:

Reviewer's Responses to Questions

**Comments to the Author**

1. Is the manuscript technically sound, and do the data support the conclusions?

Reviewer #1: Partly

Reviewer #2: Partly

Reviewer #3: Partly

2. Has the statistical analysis been performed appropriately and rigorously? 

Reviewer #1: Yes

Reviewer #2: Yes

Reviewer #3: Yes

3. Have the authors made all data underlying the findings in their manuscript fully available?

Reviewer #1: Yes

Reviewer #2: Yes

Reviewer #3: Yes

4. Is the manuscript presented in an intelligible fashion and written in standard English?

Reviewer #1: Yes

Reviewer #2: Yes

Reviewer #3: Yes

5. Review Comments to the Author

Reviewer #1: In this manuscript, Li et al demonstrated that different strains of Acinetobacter baumannii were distinct in the triggering of inflammasome activation, cell death, cytokine production, and pathogenesis. It is interesting and important to study the variation among clinical isolates affects the host’s innate response and activation of the inflammasome during A. baumannii infection. Because the antibiotic stress might cause the variation and evolution of bacterial virulence which is crucial for the pathogenesis progression.

Two major questions need to be addressed,

1. How about the bacterial growth of different strains in vitro (within culture media)? And the evidence to show these strains all belong to A. baumannii?

2. The conclusion or the correlation between the cell death and in vivo pathogenesis needs to be made more clearly during infection with different strains of Acinetobacter baumannii.

Reviewer #2: The manuscript entitled Differential activation of NLRP3 inflammasome by Acinetobacter baumannii strains by Fei-Ju Li and colleagues explores whether different Acinetobacter baumannii (Ab) strains (reference strains plus clinical isolates) trigger different levels of inflammasome activation, IL-1β release and cell death.

The authors first observe that three reference Ab strains caused different levels of lethality, bacteraemia and organ colonization in mice. They then used bone marrow-derived macrophages (BMDM) to study the expression of proteins from the inflammasome pathway. Next, they used the three reference strains plus six Ab clinical isolates to study the cleavage of the caspases. Finally, they measure the viability of cells infected with all Ab strains. They observe that the different strains show different levels of induction of IL-1β, some strains induce the release of IL-1β dependant on Caspase 11.

The objective of the manuscript is simple and it is written in a clear way. However, I think the authors have the opportunity to use their collection of strains to the best use. In a nutshell, the authors perform 4 assays:

a) qPCR on Macrophages, at three different timepoints, with 3 of 9 Ab strains.

b) Western Blot on Macrophages, at one timepoint, with all 9 Ab strains.

c) ELISA on Macrophages, at two different timepoints for 3 of 9 Ab strains; at one timepoint for all 9 Ab strains, and with two different macrophage genotypes.

d) Cell Viability on Macrophages using one assay at one timepoint for 3 of the 9 Ab strains; and in A549 cells using another assay, at one timepoint, with all 9 Ab strains.

The paper would benefit greatly if all of this would be standardized: all assays performed with all the strains at all the timepoints using both WT and Caspase-11-/- cells. This is relevant because all the Ab strains used (with the exception of reference Ab strain 17978) have roughly the same cytotoxicity ability, but might achieve this using different mechanisms. With the data showed, we have a “12h gap” between the activity of the inflammasome (Figure 3, 12h post-infection) and Cell Death (Figure 4, 24h post-infection). Finally, the authors themselves say that they “propose [a] correlation between in vitro cell death and in vitro Caspase-11-dependency.” (Lines 311 to 312), without actually showing cell death dependent on Caspase-11 on any assay.

Major comments:

- I understand that to repeat all the experiments at all the timepoints might be too much. Nonetheless, one of the two crucial experiments is to repeat the experiments performed for Figure 3 (which is at 12h post-infection) at 24 hours post-infection. Figure 2 shows that, at least at the level of mRNA, BMDMs infected with the reference Ab strains show no difference in the expression of NLRP3 or Caspase-11 at 12h post infection; but, at 24 hours post infection that strains induce different changes in the macrophages. Is it possible that cells infected with the strains change their inflammasome activation profile from 12h to 24h post infection?

- The other essential experiment is to test the cell toxicity of all the clinical isolates in macrophages. Did the different strains show different capacity to induce pyroptosis in BMDMs, as was the case for A549 cells? The authors should perform the Trypan blue and/or Zombie aqua assay in BMDMs, to assess if the higher levels of IL-1β release also correlates with increased level of pyroptosis. Also, the authors should perform the Trypan blue and/or Zombie Aqua assays in the Caspase-11-/- BMDMs if they want to conclude a death-by-Caspase 11 mechanism. (Ideally, the authors should do this at two different timepoints, but only 24h post infection is required.)

- In Figure 2a, does the Y-axis represent “Fold increase” or “Relative Expression”? I ask this because the authors draw a line at the value 1, which would mean that the graphs are showing “Relative Expression” and not “Fold increase”. This is particularly important for the interpretation of the expression of NLRC4 and AIM2 genes, since they expression is almost at 0. This can either mean slight upregulation (if the graphs show “Fold increase”) or almost no expression (if the graphs show “Relative expression”).

- Authors should quantify the results from Figure 3a. It is difficult to visually assess the levels of Caspase cleavage considering the variations seen in the GAPDH loading control band. For example, Ab isolate #916085 has the highest Caspase-1 cleavage, but also shows the highest GAPDH signal. This would also allow the authors to compile the results from all the Western blot experiments into a single graph, which would allow the reader to assess the biological variability of the activation. Finally, isolate #914394 seems to be the isolate with the highest level of Caspase-11 activation, and given that we see no full-length GSDMD, the best at cleaving Gasdermin D. So, is it possible that this isolate is the one that induces more pyroptosis?

- I would add the results from Figure S2 into Figure 4. The results shown in Figure S2 are very important for the paper, and should the presented to the reader. Also, the text refers to A594 cells being assayed with Zombie Aqua, but the Figure S2 show no such data.

- Finally, I find the abstract is written in a misleading way. “In this study, we compared nine A. baumannii strains, including clinical locally-acquired isolates, in their ability to induce activation of the inflammasome and programmed cell death pathway in primary macrophages and mice. […] Interestingly, we found a stark contrast in activation of the NLRP3 inflammasome pathway, the non-canonical caspase-11 pathway, plasmatic secretion of the proinflammatory cytokines IL-1β and IL-18 between A. baumannii strains.” While it is true the authors compare 9 Ab strains, the authors do not use the same assay for all strains, as was discussed above. This section of the abstract should be rewritten to reflect what was actually performed.

Minor comments:

- General comment for the Figures: Figure legends should be standardized. Each legend should say which statistical test was used for each panel, which cell line is being used, how many replicates for each panel (“at least three” is not precise). Figure 2 seems to have a title, whereas Figures 1, 3 and 4 don’t have a title.

- In Lines 183 to 192, which equipment was used to detect the proteins by WB?

- In Line 209, “rate” instead of “rat”.

- In Lines 210-213, there is no call for Figure 1d.

- In Figure 1a, how many mice were used to perform the survival curve?

- In the legend of Figure 1d, the authors use “canonical knockout mice” but in the text, there is no mention of these animals. So, which genotype was used to generate the results in Figure 1d?

- In Line 219, “over 24h of infection” or “for 24h of infection”, not “over for 24h of infection” as is written.

- In Line 236, first mention of “WT” in the text. Acronym should be explained, “wild type (WT)”.

- In Figure 3a, the authors should show the where the molecular weight markers are located in the blots.

- In Lines 283 to 285, the authors write: “Interestingly, neither the extended induction of Caspase-11 by the least virulent strain 17978 nor the delayed induction of NLRC4 by MDR ATCC BAA-1605 correlated with different levels of pro-inflammatory cytokines released in BMDMs.” But this seems to correlate with cell death. Can the authors elaborate on this during the discussion?

- In Lines 300 to 302, the authors write: “We performed a multiple sequence alignment of the Omp38 protein sequence between the strains ATCC 17978, 19696 and BAA 1605 and found no differences in the OmpA domain (data not shown).”. If the authors have this alignment, they should show it.

- In Line 309, “due to[space]a lack …” instead of “due to a lack …”

- The authors should detect IL-1β and TNFα by ELISA at 6h post-infection, as they have done with for the transcripts. This is the time where the transcripts show higher variability and might show differences in protein expression.

Reviewer #3: In this study, the authors demonstrated that there is a stark difference among strains of A. baumannii in inflammasome-related immune responses including pyroptosis and in vivo virulence. As well, they suggested that the correlation between in vitro cell death and caspase-11 activation can be used as a screening tool to select virulent strains in vivo, which seems to be meaningful and valuable. Follows should be checked before acceptance.

1. In Figure 3(a), cleavage of caspase-1 and GSDMD seems to be stronger in 17978-treated cells than in cells with 1605 treatment. Consistent with this, mean level of IL-1β secretion is higher in 17978-treated cells than in cells with 1605 treatment (Figure 3(b)). However, programmed cell death and caspase-11 cleavage were less in 17978-treated cells, despite of strong GSDMD cleavage. Why did it happen?

2. In legend of Figure 2, 6-24hours - 6-24 hours

6. PLOS authors have the option to publish the peer review history of their article (what does this mean?). If published, this will include your full peer review and any attached files.

Reviewer #1: No

Reviewer #2: No

Reviewer #3: No

---

## [Author Response · Author response to Decision Letter 0]

9 Sep 2022

Response to the reviewers 

5. Review Comments to the Author

Reviewer #1: In this manuscript, Li et al demonstrated that different strains of Acinetobacter baumannii were distinct in the triggering of inflammasome activation, cell death, cytokine production, and pathogenesis. It is interesting and important to study the variation among clinical isolates affects the host’s innate response and activation of the inflammasome during A. baumannii infection. Because the antibiotic stress might cause the variation and evolution of bacterial virulence which is crucial for the pathogenesis progression.

Two major questions need to be addressed,

1. How about the bacterial growth of different strains in vitro (within culture media)? And the evidence to show these strains all belong to A. baumannii?

Bacterial growth:

Response: We performed growth curve assays of these strains and have now added the growth curves into supplementary Figure 3. We added a section in the methods on the growth rate assay (L71-90) and detailed the results (L270-271). Collectively, we observed a lower growth rate for the strain ATCC 19606 strains. We found, however, no difference in growth rate between the clinical strains, or between the clinical strains and ATCC 1605 and ATCC 17978 

Acinetobacter baumannii strains?

Response: To determine whether the isolates were A. baumannii, the pathology laboratory at the Canberra Hospital performed bacterial selection on Trypticase Soy and MacConkey agar media, MALDI TOF, gram staining and colony counting. Antibiotics resistance was determine using a Vitek device (Biomerieux). We added this description in the methods section (L71-90) 

2. The conclusion or the correlation between the cell death and in vivo pathogenesis needs to be made more clearly during infection with different strains of Acinetobacter baumannii.

Response: We modified the conclusion to make our statement clearer. We stated the following (L343-L352):

In conclusion we found the MDR ATCC BAA-1605 was the most virulent strain both in vitro and in vivo and showed the highest cytotoxicity compared to our commercially available strains. Further, we also identified from our assessment of the commercially available strains that the induction of different inflammasome transcripts does not directly equate to the level of inflammation in BMDMs or cell lines. This was further supported by our assessment of clinical isolates, which showed high levels of variation in the pro-inflammatory cytokine response and programmed cell death in vitro. Additionally, due to a lack of dose-dependency in vitro, the activation of inflammasome was thought to be largely dependent on bacterial virulence. We finally propose that the combination of in vitro cell death and in vitro Caspase-11-dependency may be harnessed as a screening tool to select for more virulent strains to use in vivo.

Reviewer #2: The manuscript entitled Differential activation of NLRP3 inflammasome by Acinetobacter baumannii strains by Fei-Ju Li and colleagues explores whether different Acinetobacter baumannii (Ab) strains (reference strains plus clinical isolates) trigger different levels of inflammasome activation, IL-1β release and cell death.

The authors first observe that three reference Ab strains caused different levels of lethality, bacteraemia and organ colonization in mice. They then used bone marrow-derived macrophages (BMDM) to study the expression of proteins from the inflammasome pathway. Next, they used the three reference strains plus six Ab clinical isolates to study the cleavage of the caspases. Finally, they measure the viability of cells infected with all Ab strains. They observe that the different strains show different levels of induction of IL-1β, some strains induce the release of IL-1β dependant on Caspase 11.

The objective of the manuscript is simple and it is written in a clear way. However, I think the authors have the opportunity to use their collection of strains to the best use. In a nutshell, the authors perform 4 assays:

a) qPCR on Macrophages, at three different timepoints, with 3 of 9 Ab strains.

b) Western Blot on Macrophages, at one timepoint, with all 9 Ab strains.

c) ELISA on Macrophages, at two different timepoints for 3 of 9 Ab strains; at one timepoint for all 9 Ab strains, and with two different macrophage genotypes.

d) Cell Viability on Macrophages using one assay at one timepoint for 3 of the 9 Ab strains; and in A549 cells using another assay, at one timepoint, with all 9 Ab strains.

The paper would benefit greatly if all of this would be standardized: all assays performed with all the strains at all the timepoints using both WT and Caspase-11-/- cells. This is relevant because all the Ab strains used (with the exception of reference Ab strain 17978) have roughly the same cytotoxicity ability, but might achieve this using different mechanisms. With the data showed, we have a “12h gap” between the activity of the inflammasome (Figure 3, 12h post-infection) and Cell Death (Figure 4, 24h post-infection).

 Finally, the authors themselves say that they “propose [a] correlation between in vitro cell death and in vitro Caspase-11-dependency.” (Lines 311 to 312), without actually showing cell death dependent on Caspase-11 on any assay.

Major comments:

- I understand that to repeat all the experiments at all the timepoints might be too much. Nonetheless, one of the two crucial experiments is to repeat the experiments performed for Figure 3 (which is at 12h post-infection) at 24 hours post-infection. Figure 2 shows that, at least at the level of mRNA, BMDMs infected with the reference Ab strains show no difference in the expression of NLRP3 or Caspase-11 at 12h post infection; but, at 24 hours post infection that strains induce different changes in the macrophages. Is it possible that cells infected with the strains change their inflammasome activation profile from 12h to 24h post infection?

Response: We thank the reviewers for all these suggestions and potential additional experiments. As cell death is a likely outcome of upstream inflammasome activation we had tested earlier timepoints (4-12hr) with both LDH and Zombie aqua staining at different MOIs in BMDMs on the three commercially available strains. We found no significant differences at timepoint earlier than 24 hours (please see Supplementary data 1 and 2). We therefore retained the 24hours timepoint in our analysis. Given the lack of differences in inflammasome activation and cell death from our mRNA, LDH and Zombie aqua in BMDMs data prior to 24 hours post inoculation, we felt that performing these additional demanding experiments for all our clinical strains would not provide novel insights in reporting variation in the inflammasome response A. baumannii infection. 

- The other essential experiment is to test the cell toxicity of all the clinical isolates in macrophages. Did the different strains show different capacity to induce pyroptosis in BMDMs, as was the case for A549 cells? The authors should perform the Trypan blue and/or Zombie aqua assay in BMDMs, to assess if the higher levels of IL-1β release also correlates with increased level of pyroptosis. Also, the authors should perform the Trypan blue and/or Zombie Aqua assays in the Caspase-11-/- BMDMs if they want to conclude a death-by-Caspase 11 mechanism. (Ideally, the authors should do this at two different timepoints, but only 24h post infection is required.)

Response: We thank the reviewer for this comment. Our Western Blot data and Caspase 11 and GSDMD show that different strains have indeed the capacity of inducing pyroptosis (Figure 3). Furthermore, GSDMD activation on the Western blot data supports the activation of the pyroptosis pathway in BMDMs. Additionally the correlation between inflammasome and downstream IL-1b release is already demonstrated in Figure 3, higher level of Caspase-1 tend to show higher IL-1b release, which has been shown by previous studies (e.g. DOI:10.7150/thno.54004) to again result in GSDMD cleavage, and pyroptosis.

- In Figure 2a, does the Y-axis represent “Fold increase” or “Relative Expression”? I ask this because the authors draw a line at the value 1, which would mean that the graphs are showing “Relative Expression” and not “Fold increase”. This is particularly important for the interpretation of the expression of NLRC4 and AIM2 genes, since they expression is almost at 0. This can either mean slight upregulation (if the graphs show “Fold increase”) or almost no expression (if the graphs show “Relative expression”).

Response: The graph shows a relative expression between the strains relative to Gapdh. In the methods section we clarified the analysis methods by stating the following: “Real-time RT-PCR data and relative expression was calculated using the comparative 2-∆∆CT method with Gapdh as a housekeeping gene. Relative expression was displayed as fold-changes.” The graph shows slightly increased Nlrc4 expression and no expression of Aim2. We have updated the graphs (Y-axis labelling) in the Figure 2. 

- Authors should quantify the results from Figure 3a. It is difficult to visually assess the levels of Caspase cleavage considering the variations seen in the GAPDH loading control band. For example, Ab isolate #916085 has the highest Caspase-1 cleavage, but also shows the highest GAPDH signal. This would also allow the authors to compile the results from all the Western blot experiments into a single graph, which would allow the reader to assess the biological variability of the activation. Finally, isolate #914394 seems to be the isolate with the highest level of Caspase-11 activation, and given that we see no full-length GSDMD, the best at cleaving Gasdermin D. So, is it possible that this isolate is the one that induces more pyroptosis?

Response: We have quantified the Western blot signal and appended the results as Figure 3. Statistical significance compared to mock was reported in the figure. It was shown that isolate 914394 and 834625 were better at activating Capase-11. According to our trypan blue assay, 914394 is our fourth highest cytotoxic strain. We found it very interesting and tried to clarify this in our conclusion by saying “the induction of different inflammasome transcripts does not directly equate to the level of inflammation in BMDMs or cell lines in vitro. This was further supported by our assessment of clinical isolates, which showed a high level of variation in the pro-inflammatory cytokine response and programmed cell death in vitro”. We thank the reviewer for engaging with our data and providing this feedback to help us refine our manuscript.

- I would add the results from Figure S2 into Figure 4. The results shown in Figure S2 are very important for the paper, and should the presented to the reader. Also, the text refers to A594 cells being assayed with Zombie Aqua, but the Figure S2 show no such data.

Response: We have added the Figure S2 into the Figure 4 to present these to the readership. We removed the mention in our manuscript to Zombie Aqua in A549 cells 

- Finally, I find the abstract is written in a misleading way. “In this study, we compared nine A. baumannii strains, including clinical locally-acquired isolates, in their ability to induce activation of the inflammasome and programmed cell death pathway in primary macrophages and mice. […] Interestingly, we found a stark contrast in activation of the NLRP3 inflammasome pathway, the non-canonical caspase-11 pathway, plasmatic secretion of the proinflammatory cytokines IL-1β and IL-18 between A. baumannii strains.” While it is true the authors compare 9 Ab strains, the authors do not use the same assay for all strains, as was discussed above. This section of the abstract should be rewritten to reflect what was actually performed.

Response: We have rewritten the abstract and stated the following: “We found a variation in survival outcomes of mice and bacterial dissemination in organs among three commercially available A. baumannii strains, likely due to the differences in virulence between strains. Interestingly, we found variability among A.baumannii strain in activation of the NLRP3 inflammasome non-canonical caspase-11 pathway, plasmatic secretion of the pro-inflammatory cytokine IL-1β and programmed cell death. Our study highlights the importance of utilising multiple bacterial strains and clinical isolates with different virulence profiles to investigate the innate immune response to A. baumannii infection

Minor comments:

- General comment for the Figures: Figure legends should be standardized. Each legend should say which statistical test was used for each panel, which cell line is being used, how many replicates for each panel (“at least three” is not precise). Figure 2 seems to have a title, whereas Figures 1, 3 and 4 don’t have a title.

Response: We have updated all the Figure Legends with the reviewer’s requirements.

- In Lines 183 to 192, which equipment was used to detect the proteins by WB?

- In Line 209, “rate” instead of “rat”.

- In Lines 210-213, there is no call for Figure 1d.

- In Figure 1a, how many mice were used to perform the survival curve?

- In the legend of Figure 1d, the authors use “canonical knockout mice” but in the text, there is no mention of these animals. So, which genotype was used to generate the results in Figure 1d?

- In Line 219, “over 24h of infection” or “for 24h of infection”, not “over for 24h of infection” as is written.

- In Line 236, first mention of “WT” in the text. Acronym should be explained, “wild type (WT)”.

- In Figure 3a, the authors should show the where the molecular weight markers are located in the blots.

Response: We have updated and added these details to the manuscript 

- In Lines 283 to 285, the authors write: “Interestingly, neither the extended induction of Caspase-11 by the least virulent strain 17978 nor the delayed induction of NLRC4 by MDR ATCC BAA-1605 correlated with different levels of pro-inflammatory cytokines released in BMDMs.” But this seems to correlate with cell death. Can the authors elaborate on this during the discussion?

Response: We thank the reviewer in pointing out our mistake. We rephrase the sentence as follow: “although it does not contribute to the differential cytokine release, the extended induction of Caspase-11 by strain 17978 and the delayed induction of NLRC4 by MDR ATCC BAA-1605 correlated with cell death” 

- In Lines 300 to 302, the authors write: “We performed a multiple sequence alignment of the Omp38 protein sequence between the strains ATCC 17978, 19696 and BAA 1605 and found no differences in the OmpA domain (data not shown).”. If the authors have this alignment, they should show it.

Response: We appended the data in Suppl Figure 4 and detailed the methods (L220-225)

- In Line 309, “due to[space]a lack …” instead of “due to a lack …”

- The authors should detect IL-1β and TNFα by ELISA at 6h post-infection, as they have done with for the transcripts. This is the time where the transcripts show higher variability and might show differences in protein expression.

Response: We haven’t been able to perform a 6 hour timepoint to detect IL-1β and TNFα by ELISA.

Reviewer #3: In this study, the authors demonstrated that there is a stark difference among strains of A. baumannii in inflammasome-related immune responses including pyroptosis and in vivo virulence. As well, they suggested that the correlation between in vitro cell death and caspase-11 activation can be used as a screening tool to select virulent strains in vivo, which seems to be meaningful and valuable. Follows should be checked before acceptance.

1. In Figure 3(a), cleavage of caspase-1 and GSDMD seems to be stronger in 17978-treated cells than in cells with 1605 treatment. Consistent with this, mean level of IL-1β secretion is higher in 17978-treated cells than in cells with 1605 treatment (Figure 3(b)). However, programmed cell death and caspase-11 cleavage were less in 17978-treated cells, despite of strong GSDMD cleavage. Why did it happen?

Response: This is an interesting observation. We quantified 17978 cell death from our Western blot data. Despite the increase, we haven’t seen a significant difference. We however noted that canonical Caspase 1 activation is stronger in 17978 than other strains. It could potentially explain increase in GSDMD cleavage. 

2. In legend of Figure 2, 6-24hours - 6-24 hours

Response: We have updated corrected the manuscript and fixed the mistakes

---

## [Decision Letter · Decision Letter 1]

11 Oct 2022

PONE-D-22-16629R1Differential activation of NLRP3 inflammasome by Acinetobacter baumannii strainsPLOS ONE

Dear Dr. Burgio,

Thank you for submitting your manuscript to PLOS ONE. After careful consideration, we feel that it has merit but does not fully meet PLOS ONE’s publication criteria as it currently stands. Therefore, we invite you to submit a revised version of the manuscript that addresses the points raised during the review process.

The requirement of submitting the supplementary information may be justified.

We look forward to receiving your revised manuscript.

Kind regards,

Suresh Yenugu

Academic Editor

PLOS ONE

Journal Requirements:

Reviewers' comments:

Reviewer's Responses to Questions

**Comments to the Author**

1. If the authors have adequately addressed your comments raised in a previous round of review and you feel that this manuscript is now acceptable for publication, you may indicate that here to bypass the “Comments to the Author” section, enter your conflict of interest statement in the “Confidential to Editor” section, and submit your "Accept" recommendation.

Reviewer #1: All comments have been addressed

Reviewer #2: (No Response)

Reviewer #3: All comments have been addressed

2. Is the manuscript technically sound, and do the data support the conclusions?

Reviewer #1: Yes

Reviewer #2: Yes

Reviewer #3: Yes

3. Has the statistical analysis been performed appropriately and rigorously? 

Reviewer #1: Yes

Reviewer #2: Yes

Reviewer #3: Yes

4. Have the authors made all data underlying the findings in their manuscript fully available?

Reviewer #1: Yes

Reviewer #2: Yes

Reviewer #3: Yes

5. Is the manuscript presented in an intelligible fashion and written in standard English?

Reviewer #1: Yes

Reviewer #2: Yes

Reviewer #3: Yes

6. Review Comments to the Author

Reviewer #1: (No Response)

Reviewer #2: I raised points about analysing the available data (quantifying Western Blots) and characterizing the response at different timepoints, and the authors have adequately addressed by points and changed the manuscript accordingly.

I have two final comments:

1 – What is the purpose of the images in Supplementary Figure 1b? Are the images completely black because there is no signal? Or was there a problem in copying the images to the Word document?

2 – In Lines 323-324, authors need to correct the call to “Suppl Fig 1” to “Suppl Fig 2”.

Reviewer #3: (No Response)

7. PLOS authors have the option to publish the peer review history of their article (what does this mean?). If published, this will include your full peer review and any attached files.

Reviewer #1: No

Reviewer #2: No

Reviewer #3: No

---

## [Author Response · Author response to Decision Letter 1]

13 Oct 2022

Reviewer #1: (No Response)

Reviewer #2: I raised points about analysing the available data (quantifying Western Blots) and characterizing the response at different timepoints, and the authors have adequately addressed by points and changed the manuscript accordingly.

I have two final comments:

1 – What is the purpose of the images in Supplementary Figure 1b? Are the images completely black because there is no signal? Or was there a problem in copying the images to the Word document?

Response: The mages in the supplementary data provide a cell death quantification by Zombie Aqua from 6 to 24 hours post inoculation. The images are representative field for quantification. We do apologise for the inconvenience. It seems the image conversion renders the images black. In the revised version, we have enhanced the contrast of the images to ensure these are not black. 

2 – In Lines 323-324, authors need to correct the call to “Suppl Fig 1” to “Suppl Fig 2”.

Response: Many thanks for spotting this mistake. We have now fixed it.

Reviewer #3: (No Response)

---

## [Decision Letter · Decision Letter 2]

18 Oct 2022

Differential activation of NLRP3 inflammasome by Acinetobacter baumannii strains

PONE-D-22-16629R2

Dear Dr. Burgio,

We’re pleased to inform you that your manuscript has been judged scientifically suitable for publication and will be formally accepted for publication once it meets all outstanding technical requirements.

Kind regards,

Suresh Yenugu

Academic Editor

PLOS ONE

Additional Editor Comments (optional):

Reviewers' comments:

Reviewer's Responses to Questions

**Comments to the Author**

1. If the authors have adequately addressed your comments raised in a previous round of review and you feel that this manuscript is now acceptable for publication, you may indicate that here to bypass the “Comments to the Author” section, enter your conflict of interest statement in the “Confidential to Editor” section, and submit your "Accept" recommendation.

Reviewer #2: All comments have been addressed

2. Is the manuscript technically sound, and do the data support the conclusions?

Reviewer #2: Yes

3. Has the statistical analysis been performed appropriately and rigorously? 

Reviewer #2: Yes

4. Have the authors made all data underlying the findings in their manuscript fully available?

Reviewer #2: Yes

5. Is the manuscript presented in an intelligible fashion and written in standard English?

Reviewer #2: Yes

6. Review Comments to the Author

Reviewer #2: no further comments. The authors have corrected the few minor technical issues and the manuscriptis now acceptable for publication.

7. PLOS authors have the option to publish the peer review history of their article (what does this mean?). If published, this will include your full peer review and any attached files.

Reviewer #2: No

---

## [Editor Report · Acceptance letter]

24 Oct 2022

PONE-D-22-16629R2 

Differential activation of NLRP3 inflammasome by *Acinetobacter baumannii* strains 

Dear Dr. Burgio:

I'm pleased to inform you that your manuscript has been deemed suitable for publication in PLOS ONE. Congratulations! Your manuscript is now with our production department. 

Kind regards, 

on behalf of

Dr. Suresh Yenugu 

Academic Editor

PLOS ONE